Original research

# Factors that influence patient preferences for virtual consultations in an orthopaedic rehabilitation setting: a qualitative study

Anthony W Gilbert ,[1,2,3] Jeremy Jones,[2] Maria Stokes ,[2,4] Carl R May [3,5]

[1]Therapies Department, Royal National Orthopaedic Hospital Stanmore, Stanmore, UK
[2]Faculty of Health Sciences, University of Southampton, Southampton, UK
[3]NIHR Applied Research Collaboration, North Thames, UK
[4]NIHR Applied Research Collaboration, Wessex, UK
[5]Faculty of Public Health and Policy, London School of Hygiene and Tropical Medicine, London, UK

**Correspondence to**
Anthony W Gilbert;
anthony.gilbert@nhs.net

## ABSTRACT

**Objectives** To identify, characterise and explain factors that influence patient preferences, from the perspective of patients and clinicians, for virtual consultations in an orthopaedic rehabilitation setting.

**Design** Qualitative study using semi-structured interviews and abductive analysis.

**Setting** A physiotherapy and occupational therapy department situated within a tertiary orthopaedic centre in the UK.

**Participants** Patients who were receiving orthopaedic rehabilitation for a musculoskeletal problem. Occupational therapists, physiotherapists or therapy technicians involved in the delivery of orthopaedic rehabilitation for patients with a musculoskeletal problem.

**Results** Twenty-two patients and 22 healthcare professionals were interviewed. The average interview length was 48 minutes. Four major factors were found to influence preference: the situation of care (the ways that patients understand and explain their clinical status, their treatment requirements and the care pathway), the expectations of care (influenced by a patients desire for contact, psychological status, previous care and perceived requirements), the demands on the patient (due to each patients respective social situation and the consequences of choice) and the capacity to allocate resources to care (these include financial, infrastructural, social and healthcare resources).

**Conclusion** This study has identified key factors that appear to influence patient preference for virtual consultations in orthopaedic rehabilitation. A conceptual model of these factors, derived from empirical data, has been developed highlighting how they combine and compete. A series of questions, based on these factors, have been developed to support identification of preferences in a clinical setting.

## BACKGROUND

Videoconferencing technologies, such as Skype, Zoom, Attend Anywhere and Microsoft Teams, have been received enthusiastically by healthcare policy makers[1–3] as they provide a medium to improve access to care. The technology is also viewed as a significant contributor to health and wealth[4] and efficiency gain strategies.[5] Videoconferencing technologies

### Strengths and limitations of this study

► This is the first qualitative investigation of patient preferences for virtual consultation in a tertiary orthopaedic setting.
► Theoretical insights and explanations generated from this paper are developed from empirical data.
► Maximum variation sampling and abductive qualitative analysis reveal key factors that shape patient preferences.
► Single site qualitative study is not generalisable but mechanistic model is likely to be transportable between settings.

are being used across many fields of healthcare[6] and can offer advantages to patients. In January 2020, the UK recorded its first case of Novel Coronavirus (COVID-19). The outbreak of COVID-19 placed the NHS (National Health Service) under significant strain. Social distancing measures were introduced in the UK in March 2020 and virtual consultations (VC) (via telephone or video call) were identified as a potential alternative to face-to-face consultations at this time.[7 8] Organisations were forced to rapidly implement VC as a consequence of COVID-19.[9]

Greenhalgh *et al*[10] conducted a multilevel mixed methods study of Skype consultations and found that they were safe, effective and convenient for patients when healthcare professionals judged them clinically appropriate. However, the authors[10] found that the reality of establishing VCs in outpatient services was more complex than originally anticipated. This complexity is a longstanding problem in the implementation of telemedicine and telecare systems.[11]

### Patient preferences and burden of treatment

A preference can be defined as an individualised 'total subjective comparative evaluation'.[12] Put simply, an individual weighs up the

characteristics of alternatives to make a decision. Preference theory suggests that a person will prefer the outcome that yields greatest utility, and therefore that patients would prefer a VC if they believe its benefits outweigh its burdens.[12] To date, patient preferences for telemedicine have only been investigated at a general population level.[13]

VCs have been shown to change what is required of patients.[14–16] A workload for patients that exceeds their capacity has been demonstrated to be a driver of treatment burden for those with lung cancer and chronic obstructive pulmonary disease.[17] Treatment burden in patients with stroke has been shown to be influenced by the quality and configurations of healthcare.[18] What is not yet understood is how changes in the work and demands of being a patient as a result of VC influence preference for VC in a healthcare setting.

Patients' and professionals' preferences for telemedicine are not isolated from their other experiences of healthcare, or from the ways that they experience other aspects of their lives. If we are interested in the ways that patients understand and calculate the relationship between benefits and burdens, then we should also include burdens in our investigation. Shippee *et al*'s[19] cumulative complexity model assumes an arithmetical relationship between delegated health system workload and individual patient capacity, and suggests that this explains healthcare utilisation. However, health behaviours and service utilisation take place in a broader social context, and burden of treatment (BoT) theory [20] provides a way into this problem. BoT explains the relationship between the demands that participating in healthcare places on patients and caregivers (their workload), and the affective, cognitive, relational and material resources that they can bring to bear on this workload (their capacity).[21 22]

To our knowledge, no studies have yet investigated the relationship between patient preferences around telemedicine services and their experience of burden of treatment. We need to better understand this to support the development of care pathways that take into account what offers patients increased utility. This paper therefore aims to identify, characterise and explain factors that influence patient preferences for VCs in an orthopaedic rehabilitation setting.

## METHODS

This paper is part of a larger body of work and forms phase II of the CONNECT Project (Care in Orthopaedics, burdeN of treatmeNt and the Effect of Communication Technology). The protocol for the CONNECT Project has been published elsewhere.[23]

### Setting

The research was conducted within a single specialist orthopaedic hospital in North London, UK. All participants were recruited from the Occupational Therapy and Physiotherapy Department.

### Participants

A maximum variation sample was recruited; we intended to sample our patients on a set criteria of variation (set for age and gender for patients and occupation for clinicians). This included 22 patients and 22 healthcare professionals (see table 1 for the inclusion and exclusion criteria). We aimed to recruit at least 10 male and 10 female patients (10<50 years and 10>50 years) and 20 healthcare professionals (occupational therapists and physiotherapists). Patients were selected to be interviewed to identify factors that influence patient preferences for VCs. Clinicians were selected to be interviewed to provide their perspectives on patient preference and as patient preferences are moderated by the possibilities and preferences of organisations and staff. The first two patients and healthcare professionals were used to pilot the interview schedule (see online supplemental materials 1 and 2).

### Recruitment

The study was advertised using a pop-up banner in the Occupational Therapy and Physiotherapy Departments. Patients were encouraged to discuss the study with their treating healthcare professional or could approach the researcher directly via email. Healthcare professionals were sent a departmental wide email informing them of the study both from the perspective of discussing with patients as well as enrolling as a participant. Suitable and interested potential participants were provided with a participant information sheet and given at least 24 hours to discuss the study with

| Table 1 | Inclusion and exclusion criteria |
| --- | --- |
| **Inclusion criteria** | **Exclusion criteria** |
| ► Patients, over the age of 18 years, attending the host institution for Physiotherapy or Occupational Therapy<br>► Patients who have experience of orthopaedic / musculoskeletal condition<br>► Patients who are able to provide informed written consent to enter into the study<br>► Patients able to understand and speak English or a language covered by the host institution interpreter service<br>► Physiotherapists or Occupational Therapists (or assistants) who treat patients with orthopaedic / musculoskeletal disorders | ► Patients without the capacity to consent<br>► Patients suffering from disorders other than orthopaedic as the primary cause (eg, neurological or oncology disorders)<br>► Physiotherapists or Occupational Therapists who do not currently treat, or have no experience of treating patients with orthopaedic / musculoskeletal disorders<br>► Patients currently or previously treated by AWG |

the researcher. They were enrolled in the study on receipt of informed written consent.

## Data collection

Design of the interview schedules were formed by BoT theory[24] (see online supplemental materials 1 and 2) and the results of Phase I of the CONNECT Project.[15] Interviews were conducted on-site at the hospital or virtually using phone or Skype. Interviews were conducted by AWG and were to last around 60 min with the option to extend or shorten as required. All interviews were audio recorded and sent off for transcription to an external company. All transcripts were emailed or posted to participants on receipt to give them the option to verify the data or to make any adjustments.

## Data analysis

Interview transcripts were reviewed and uploaded into NVivo (V.12). Data analysis followed the principles of abduction as set out by Tavory and Timmermans.[25] Coding was undertaken by AWG and CRM. Open coding techniques were used to identify empirical regularities (themes) in the data. Data that matched the results of the CONNECT Project Phase I were temporarily set aside; this research sought abductive 'surprises' (new themes) in additions to those gained from our previous work. Useful data to support the design of a Discrete Choice Experiment (a forthcoming paper that constitutes phase III of the CONNECT Project) were set aside. The new themes were interrogated for attributions about patient preferences and the factors that shape them. Attributions were assigned to codes within these new themes following discussion between AWG and CRM. Attributions were subsequently discussed between AWG and JJ to ensure they made sense and were accurate representations of these data. No changes were required to attributions at this stage. Inferences were made about the ways that preferences worked, the relative position and significance of the factors that shaped them, forming abductive explanation. Data matching the themes from Phase I were then incorporated

once theoretical insights were formed. Finally, themes arising from the data were mapped out in a model by AWG to visualise how different factors might influence preference for virtual consultations. The theoretical model was reviewed by all authors to verify its content. A summary of these methods can be seen in figure 1. Reporting was conducted using the Standards for Reporting Qualitative Research[26] (see online supplemental material 3).

## Patient and public involvement

The CONNECT Project patient and public involvement steering group (PPISG) has been set up to provide guidance on the conduct of the research (details available from www.theconnectproject.info). The first meeting of the PPISG was held in August 2016 prior to the submission of the research to the National Institute for Health Research in May 2017. A discussion was held about the overall research aims which supported the identification of the research questions. The PPISG has supported the design of the overall research plan and will continue to be involved during the development and refinement of each phase prior to the completion of each study protocol. The participant information and consent forms and the discussion guide for this research was reviewed by the PPISG. In addition, the PPISG will support the development of the lay summary outputs to be disseminated to patients and the public.

## RESULTS

No changes were made to the interview schedule after the pilot interviews and these data were included in the study. Fourty-four participants were interviewed in the study; 22 patients (12 women, average age 46 (range 20–78)) and 22 healthcare professionals (13 physiotherapists, 14 women). The average interview length was 48 minutes (range 28–81 minutes). Two patient interviews were conducted over the phone and two over Skype. Two healthcare professional interviews were conducted over

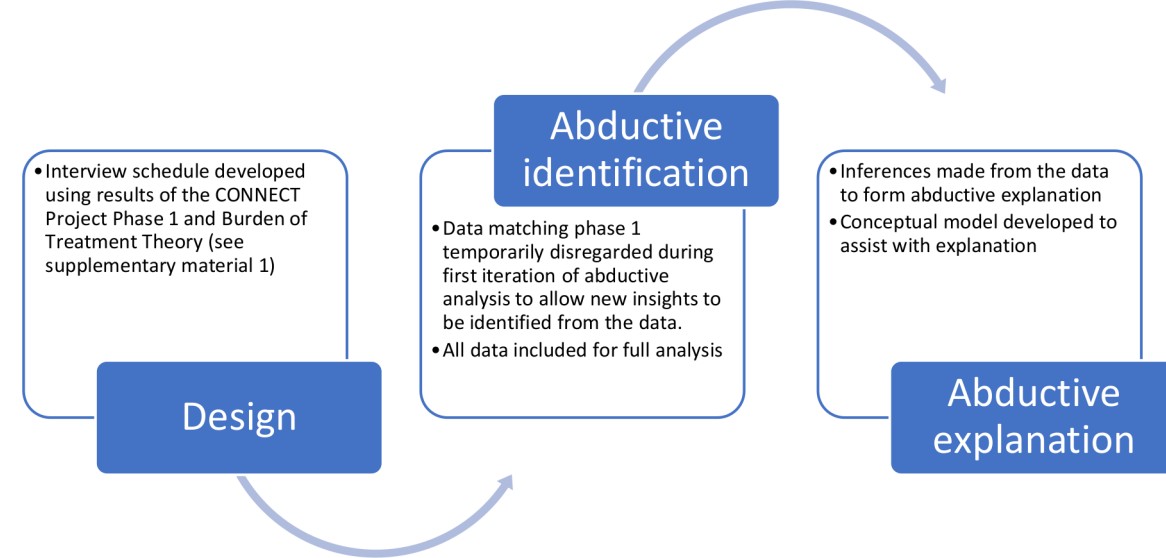

**Figure 1** Flow diagram of methods.

**Table 2** Theme 1: situation of care

| Factor | Description | Patients accounts | Healthcare professionals accounts |
|---|---|---|---|
| Clinical status | The healthcare complaint the patient experiences, its stability, reversibility and its impact on the patient in conjunction with other complaints. | If I'm having a flare-up, sometimes I can't even leave the house. I get stuck indoors and I just wouldn't be able to do much really (P7) It was really annoying because it had, like, dislocated, it was dislocated loads before and after to the point that it was really affecting my life. Then I got banned from doing stairs, I couldn't go out here, I couldn't go out there, couldn't really walk anywhere (P5) | You go back, and then sometimes they make an x amount of improvement, or they have a flare up and then it goes back a bit because they get really stressed out. They're back to that fearful of movement (C7) They're not managing those flare-ups particularly well, so they end up missing classes and things like that. It's become a bit of a spiral to have that—the physical is having a knock on the mental which is having a knock-on effect on the physical and they're just spiralling out of control (C14) |
| Treatment requirements | The treatment and management of the complaint that is required. The restrictions imposed on the patient. | But after surgery, I was literally bedbound for 3 months, so for 3 months I couldn't do anything (P20) We're just building up my stamina I think at the moment. Not with the hands but with the shoulders. We're just starting slow, building up (P3) So, they've basically come up with a programme for my gym telling me how often I should do it, giving me encouragement saying you're a bit better (P6) | … building arm strength, stability, muscle patterning, working whole kinetic chain, core stability, lots and lots of gluteal rehab, putting a big emphasis on to their understanding of what's a good muscle ache and what they should be feeling and what's working to fatigue rather than what's working into their pain, and then understanding what's an okay pain to have, what's okay to work through, what's not okay to work through (C11) |
| Care pathway | The availability of healthcare to the patient | On a Skype, are you going to have a half an hour appointment? Or are you just—is it just a check-up to see that you're doing the exercises correctly and they say, right, okay, fine carry on with those? Or that looks really good. So, I think it depends on the time apart, how far you are from the hospital (P2) So if it was once every 3 months, I'd definitely prefer to have—and so, maybe the later stages and everything's better, then I wouldn't mind having the Skype session, but in terms of the actual rehab and getting from surgery back to performance, I'd definitely like to see a physio (P20) | …face-to-face slots for me particularly can be—would be really normal to have to wait 6 to 8 weeks for another appointment just because of our system and the vast amount of patients that we have (C15) I think doing it as an adjunct where it's extra, we just don't have the capacity for a start, even if it was to (text doing), doing things like that. I think that would be difficult to fit in (C1) At the moment our face-to-faces are an hour. We don't know that when we do virtual it could be actually much more efficient for us. We could do a really good 30 min telephone consultation and we can actually fit more of them in (C18) |

the phone. No participants returned their transcripts and therefore no amendments were made.

### Interview data

Four themes were identified from the data: (1) the situation of care, (2) expectations of care, (3) demands on the patient and (4) capacity to allocate resources to care. Results from interviews are presented by theme and evidenced in tables 2–5 which present data from both patients and healthcare professionals.

### Theme 1: situation of care

The situation represents the ways that patients understand and explain their clinical status, their treatment requirements and the care pathway.

### Clinical status

Patient preferences varied based on the clinical challenges patients faced at that time and the patient's capacity to meet the demands the clinical status required. Healthcare professionals had an awareness of the volatile nature

**Table 3** Theme 2: expectations of care

| Factor | Description | Patients accounts | Healthcare professionals accounts |
|---|---|---|---|
| Desire for contact | Whether the patient / healthcare professional believes the F2F is more of a capable method of care delivery than VC. | I'm sure I could do that at home on my own but personally I would feel comfortable knowing I've got a person actually feeling it(P16)<br>If it's something simple then, yes, that's a good idea. If it's something a bit more complicated they actually have to come and see it because it's more of a hands-on type of thing (P8) | We definitely can't do is gait re-education or gait analysis. We could probably demonstrate exercises ourselves, but actually if we're looking at a movement habit in terms of, say, how someone's shoulder moves, or you need to really see or perhaps feel what that is, I think that's obviously not able to do that (C15)<br>Obviously, if it was a more physical session, if it was a practical session, that's not going to work particularly well; it's not going to work very well on Skype (C12) |
| Psychological status | The psychological status of the patient and the impact of this on care across different delivery formats. | One of the reasons why the screens would be good is I would feel less anxious to talk to someone through a screen, but I would in the same room (P9)<br>You don't like the way that your life's going to look because you know you're not going to be able to achieve all the things that you want to achieve (P17)<br>Over the years I have done a lot of leg and knee exercises… especially immediately after surgery… I probably should keep them going but I have to say I haven't (PP2)<br>I guess because I was in a leg brace for so long, stuff did get shouted at me and I did get called things and that, so my self-confidence isn't the best in the world(…) So to see myself in the corner of a screen doing something, it would stress me out for quite a huge amount of time (P5) | It might also make them feel a bit less anxious about having to travel, having to worry if my therapist or whoever I'm coming to see makes me feel welcome or makes me feel comfortable… It might make them feel a bit more comfortable if they're in their own environment (C16)<br>I think it's that how much do the patients value that just talking to someone in person, that relationship side of things and those sorts of things that maybe they might not feel so safe to do … and also sometimes patients just want a hug (C1) |
| Previous care | Experience of previous care | Yeah, I think you, for me, I feel like I've been able to build up more of a bond with them all because I've seen them in person, whereas if it had been over a screen or a phone, I don't think I would have had that (P5)<br>So, I've had physio on and off for fibromyalgia and actually I've been able to connect with this much better because of the way it's delivered (P3) | I don't think you can give a one size fits all to people. Some men particularly they just want a number, they want a number, they want sets they want reps. They just want a very clear structure and some people just you have to go that way because they react better to it. They're more likely to be more adherent to exercise if they go that way. Other people it's just a case of listening to your body, see how you feel, see what you manage. Because if you push them too far or push too little you could—you're just going to end up failing them, I think (C14) |
| Perceived requirements | The negotiated requirements of the session | 'We tend to come down to [the host institution] probably once every 6 months now just for a check-in… so that she can then check-up on those joints and make sure that I don't need to change what I'm doing or we don't need to look into it and get things investigated with orthopaedics (P17) | I think it also depends on the population. Not everyone has complex needs as well. I think if we have a routine primary knee replacement there's no reason why you can't get everything. If you have a flare referral you'd be fine to do a 30 min, whereas if you have a revision who's had five surgeries, 30 min is probably not going to be enough, because there will be a lot of belief systems around that which probably need to be looked into. So, yes and no. It depends on what the patient group is (C7) |

F2F, face-to-face; RNOH, Royal National Orthopaedic Hospital; VC, virtual consultation.

of patient's clinical status. Patients who had a long-term orthopaedic condition had an awareness that their clinical status has the potential to both worsen and improve with some patients experiencing this degree of volatility. The patient's orthopaedic problem could stand alone or be in conjunction with other physical or mental health issues.

### Treatment requirements

The requirements of treatment are dependent on the clinical status of the patient, in accordance with the normal management for that status. A spectrum of management strategies may be required, some of which traditionally require hands-on treatment and others which can be delivered without physical contact. Some clinical status' require forced restriction of activities which make physical attendance challenging, whereas other status' require physical contact.

### Care pathway

Patient preferences are influenced by the care that is available. This includes the length of the appointment, number of appointments and regularity of these and

**Table 4** Theme 3: demands on the patient

| Factor | Description | Patients accounts | Healthcare professionals accounts |
|---|---|---|---|
| Care requirements | The requirements of care | It depends what you're asking them to—if it was—it depends. If it's something simple then, yes, that's a good idea. If it's something a bit more complicated they actually have to come and see it because it's more of a hands-on type of thing (P8)<br><br>I suppose it's not so much the conversations but the physical things that you might have to do. It would be very difficult for them to work out—if you're talking physiotherapy—just how your joints were working. They couldn't really see what your back was doing or how your arm was working or whatever, and you can't—they need to feel. Physiotherapy's quite a hands on the body sort of thing (P4)<br><br>It's ridiculous in the sense that appointments have almost become a full-time job for me. I'm really grateful, I've got a lovely team of people that know me very well and look after me (P10) | How many exercises can they realistically fit in their day? I'd rather they did one or two really well then five or six badly (C11)<br><br>I guess if they've had no restrictions really at all, then to completely have those restrictions—and it can be quite debilitating because they're so used to being independent and not having to really rely on others (C4)<br><br>We do often use our hands for some assessment in terms of feeling for muscle-activated patterns or guarding (C15)<br><br>We do lay on our hands. It might well be around showing someone that they've become really hypersensitive. Touching them on an area of skin that is not at all uncomfortable and saying what does that feel like, does it feel like I'm poking, whatever, and then putting your hand on their back or something and then say how does that feel? (C10) |
| Social demands | The competing life demands that can interfere with healthcare. | I think, because I'm not looking after my mum, my mum has gone into a care home now. At the moment I haven't a job. I'm not working. I'm at home, I'm just doing things at home. I still go to the care home and sort things out for mum and appointments and that (P2) | I think for some people things are muddling along and I probably should work on my routine, but I've got my kids, I've got my work—this takes priority and that's I think my role is trying to tease that out a bit more. So, what is your priority right now? (C12)<br><br>Maybe this is where the overwhelmingness comes in because if you are not doing any of things you suddenly feel like you have to change your entire life to be able to manage if some of what we have said isn't said carefully (PC1) |
| Consequences of choice | The impact of choice | For me, it's the equipment. I only live in a small—and it is small, isn't it—a small two-bedroom house. I would have nowhere to store the equipment… there's no option out there to rent equipment (P19)<br><br>Some of the stuff he doesn't need to touch me for, like when he's watching me do a squat. Are my knees going the right way? Yeah. He can do that over a FaceTime. That's absolutely fine. But as you say, he needs to—if he wants to check my strength physically, then yeah, I need to be here. It only limits that (P14) | You might subconsciously use that (travel time) in a beneficial way… If you are straight in on a computer screen maybe there is some prep time that is not build in to the process as easily and you have to be mindful of preparing yourself beforehand (PC1)<br><br>*If you think about the patient that is actually sent into a flare-up from the journey that they've made…(C8)*<br><br>So often if they want to try and demonstrate exercises, a common feedback is the fact that their bed's too hard or too soft and it doesn't work, and the plinths are easier to do it (C1) |

the time of day of the appointments. Some patients who found accessing care challenging would feel less inclined to travel if the appointment was very short or at an inconvenient time of day. Others would be prepared to travel, whatever the offering. Regular repeated appointments can be burdensome for patients, particularly those with other commitments that might use up capacity. Patients with infrequent appointments appeared to favour

**Table 5** Theme 4: capacity to allocate resources to care

| Factor | Description | Patients accounts | Healthcare professionals accounts |
|---|---|---|---|
| Financial | The ability to free up financial resources | So obviously taking an afternoon off as annual leave or whatever wouldn't result in a disciplinary, but then in the long-run I have to think…(P5)<br>If you're doing it once a week or something, you're spacing it out… it's travelling there. That would be—it's expensive to travel up here because it's not exactly in the closest of areas, it's in the middle of nowhere (P7) | They might have a bit more support but again they've then got to think about to do—if they're paying for it privately there's the added cost to them (C4)<br>When I think about some of these patients that come like 3 hours on public transport—what a waste of money that is. I think of patients that come all the way from Birmingham and Brighton. That doesn't make any sense to me, and actually at times I have said I think we should do this on the phone (C17) |
| Infrastructure | Access to material and informational resources | You could get a stand and you'd be able to see everything really. If you put it on a table, if you need to sit on a chair. You could pull it a bit away from you so they can see you. I reckon definitely it would work (P7)<br>I would either Skype on my laptop or Skype on my thing, and if I could transfer to the TV, you know? I've got a smart TV, it could be done that way. Because if you've got a bigger picture you could see more, you could do more, whereas if you've got a little screen your vision is very limited to a little square (P8) | If you haven't got a laptop and Skype at home, then you're probably not going to be that techy, that tech savvy, and that open to learning how to use a tablet that you've never used before or something, probably (C19)<br>They would need access to the technology… do they have the Internet, do they have a connection, do they have a smart device, do they have a way that they can use that and are they familiar with their platform… a prime example is Skype. iPhone users tend to use FaceTime so do they have a Skype account, are they able to set it up? I think it's that accessibility, and it's have they used it before which is a big thing…(PC2) |
| Social capacity | Support available through social network | I have a husband who does lots of stuff for me… I can't do housework because I can't lift an iron anymore (P4)<br>Without that group, I think I would just be in bits right now to be honest.(P14) | This woman, who I was talking about just before, she lived by herself and she hasn't got any carers but the family was helping (C2)<br>More patients are having their family members helping them with these things at home and that visit regularly. There's no reason why that can't be—if they're turning up to help them put on TED stockings, then I'm sure they can help them turn on a tablet and watch something (C5) |
| Healthcare system | Sources of healthcare capacity | I think it's emotional support as well. I suppose in my case because I've had so many mental issues attached to my disorder, I have found support here from an orthopaedic point of view. When I had a setback and I was told there was a potential another infection in my bone I went to pieces here, and I saw (anonymised). He was so reassuring… I know I've got security because I feel (anonymised) knows my case so well, and he knows what happened (P10)<br>It's difficult for me, I can't use the underground or anything like that so I use the patient transport and they fetch me… some of those appointments have been 10 min or so and I have used the patient transport…(PP2) | But the skill then is to watch your language and rather than tell someone how easy it is, or tell someone the solution, again that's where motivational interviewing comes in. Rather than saying but you can just pace, let's work out how you can pace, say something like is there anything that you've been learning that you feel could give some boundaries there or anything you've tried? So again, you're getting the person to solve their own problems (C13)<br>Sometimes the hospital transports are not quite helpful for them. They don't come on time, so they delay sometimes. She ends up missing her appointment because of a delay in the hospital transport (C2) |

face-to-face (F2F) appointments, although there were exceptions to this. Healthcare professionals commented on the rigidity of corporate resources, with some finding the volume of workload reduced their capacity to be flexible, for instance finding time to support patients with managing their VC.

### Theme 2: expectations of care

Patients have expectations for both VC and F2F consultations. These expectations are influenced by a patient's desire for contact, psychological status, previous care and perceived requirements.

### Desire for contact

Patients had beliefs about the effectiveness of a VCs in comparison to a F2F therapy session. They preferred F2F consultations if they believed they would have more favourable outcomes as a result. Patients also preferred F2F contact if they felt their condition was complicated and warranted a physical examination. Healthcare professionals believed that VCs were not capable of delivering the physical aspect of a session.

### Psychological status

Patient motivation and self-efficacy was an important consideration for both patients and healthcare professionals. Some patients felt they were less likely to complete prescribed care if they were attending virtually whereas others felt that VCs could reduce the anxieties associated with F2F interactions and travelling into the hospital. Some patients, however, found the idea of seeing

themselves on a screen stressful. Healthcare professionals had an awareness of the potential limitations to offer empathy via VC to the patients who desired it.

### Previous care

Patients previous experience influenced their preference for VC. Patients who had built up a good rapport with their current care team felt that they want F2F to continue whereas others felt that, as they trusted their healthcare professionals, they would be willing to try a new innovation. Patients who had received suboptimal care elsewhere felt that they would be more likely to stick to the status quo if this worked well for them. Healthcare professionals were sensitive to the varied experiences and expectation of patients.

### Perceived requirements

Patients who feel the need for hands on F2F care reported a preference towards F2F care. Patients who did not feel F2F was necessary did not feel the same way. Care requirements differed based on the individual circumstances of the patient and the length of time of the appointment. Patients who travelled less frequently preferred to receive a physical examination, often as a 'check-up' to assess the physical status of the problem.

### Theme 3: demands on the patient

Patients may face multiple and differing demands dependent on the choices they make regarding a VC or a F2F consultation. Demands include the care requirements, social demands and the consequences of choice.

### Care requirements

The care requirements are dependent on the clinical status of the patient. Patients may be required to complete complex exercise regimens or perform assessments. Some of these initiatives may benefit from optimal visualisation of movements. Some of these may require hands on facilitation. For others, manual therapy may be indicated. Preferences are likely to be mediated by what the healthcare professional believes and the consequence of choice will change the demands on patients. These changes may be burdensome depending on the patient's capacity.

### Social demands

Some patients in this study reported a vast array of social demands that interfered with healthcare, such as caring for elderly relatives or young children. Often, these conflicting demands interfered with the patient's ability to attend their own appointments and rehabilitation. Patients who reported excessive social demands reported that in some circumstances VCs could be more favourable.

### Consequence of choice

The use of virtual consultation equipment may require a new skill set. Patients might also need to obtain rehabilitation equipment and technology for VC. Patients who did not have the space and rehabilitation equipment available preferred to travel in for a F2F consultation. Patients

that found the idea of interacting with their rehabilitation professional over a screen challenging where more likely to prefer F2F appointments whereas others did not see this as an issue. Overcoming the lack of physical contact and adapting assessments proved to be an issue for some. The lack of a suitable rehab environment was a concern to some healthcare professionals.

The demands faced by patients arose as a direct result of the situation in conjunction with the capacity to fulfil the demands. Patients who felt that VCs were less burdensome may have a preference towards VCs whereas those who find them more burdensome may have a preference towards F2F consultations.

### Theme 4: capacity to allocate resources to care

*Capacity* is the patient's ability to allocate resources to care. These resources are financial, infrastructural, social and healthcare related.

### Financial

Patients found that the demands of travel to a physical appointment can be costly, particularly when this entailed long journeys by public transport. Some patients were required to take unpaid leave from employment or risk losing their job. Some patients had supportive employers or did not feel significantly impacted through the cost of attendance. Healthcare professionals were aware of these financial challenges faced by patients.

### Infrastructure

Patients needed to have access to the hardware and software in order to use VC as a form of consultation. There was a requirement to understand how to use the technology in order to undergo a successful VC. Variations of hardware and software exist. There did not appear to be any relationship with type of hardware and software combination and preference. Some devices with larger screens were thought to be more beneficial and influence *expectations*. In addition, patients needed to have access to a suitable environment and equipment in order to undergo virtual rehabilitation.

### Social capacity

Patients who had a support network available to them found this was a useful resource. Family members were able to assist with the logistics of travel to appointments, activities routines at home and motivation to engage with rehabilitation programmes. Healthcare professionals reported ways in which patients could enhance capacity through their social networks.

### Healthcare system

The healthcare system can provide capacity. For example, some patients received hospital-funded transport making attendance at the hospital easier. Healthcare professionals are skilled at facilitating motivation and behaviour change which could improve capacity. Expectations of success may provide patients with additional motivation and self-efficacy to achieve the demands required of them.

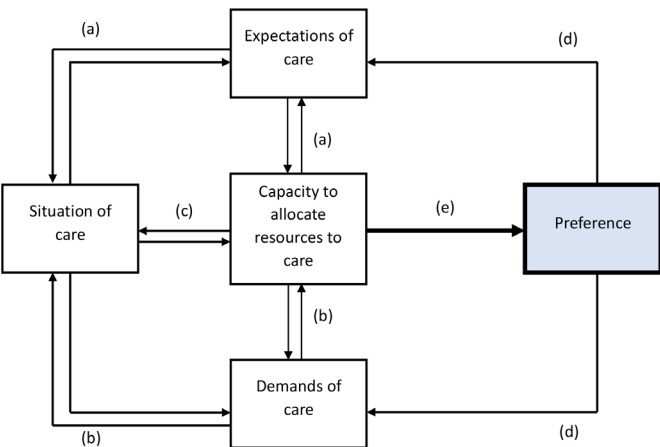

**Figure 2** Model to illustrate interactions between mechanisms that influence preference for virtual consultations.

Capacity is an important mediator of preference as it dictates whether or not a patient has the available resources to meet the demands of the situation and the expectations. Capacity is a mediator between the types of influences at work and has a direct influence on preference (see figure 2).

The *Situation* is a factor that influences preference. Each situation is unique to the individual based on their clinical status, treatment requirements and the availability of care. The situation is influenced by the *Capacity* of the patient to allocate resources to care which in turn influences the *Demands* and the *Expectations of* patients. While certain factors influence preferences for a patient in one direction, other factors may have an opposite effect.

## DISCUSSION

This paper outlines four key factors and describes mechanisms that influence patient preferences in the context of VC for orthopaedic rehabilitation. These factors have been empirically derived. These factors have been identified and characterised, and can be mapped as an explanatory model that demonstrates the interplay between factors and how they interact to influence preferences.

(a) The relationship between *Situation of care* and *Expectations of care*

The situation informs the patient's expectations of care. If the situation demands F2F (or VC) the patient will be required to decide whether F2F (or VC) would be the most suitable alternative based on the care they expect to receive. These expectations influences the situation of care for the patient.

(b) The relationship between *Situation of care* and *Demands of care*

The situation requires the patient to perform specific tasks to engage in their care. These demands will fluctuate as the clinical status and the treatment requirements fluctuate. The availability of the care pathway may remain fixed or fluid dependent on the specific situation. Resources available through capacity will dictate the

demands of the situation. Competing demands on the patient may reduce available capacity to complete the demands of care dictated by the situation. The demands on the patient, and their interaction with the patient's capacity in turn influences the situation.

(c) The relationship between *Situation of care* and *Capacity to allocate resources to care*

Patient capacity influences patient expectations indirectly via the demands and expectations of care. In addition; the capacity of the patient to engage with care itself can influence the situation as resources may be allocated to the patient by the healthcare provider depending on a need's basis, for example, whether a patient qualifies for hospital-funded transport. The capacity of the patient to engage with care is therefore directly dependent on the situation.

(d) The consequences of *Preference*

The preferred choice between a F2F and a VC has consequences. The consequences of choice directly impact on the demands of the patient and their expectations of care. Changes in expectations and demand in turn influence the patient's capacity and the situation.

(e) The formation of *Preference*

The formation of preference, within this study, is the resulting process of complex factors interacting with one another. The establishment of the situation and capacity dictate the expectations and demands of care. Preferences are established following a total (considering the options available) subjective comparative (these options are compared based on the patient's experience) evaluation (the option with the most utility is selected).

A total subjective comparative evaluation is a cognitively demanding task.[12] We have found, from this research that multiple factors are at play that combine and compete. To ask sensitising questions in relation to these factors may facilitate the cognitively demanding task of preference formation. These results can therefore be applied to clinical care in the form of sensitising questions for clinicians to ask patients to support formation of preferences for or against F2F (or VC). These questions have been developed from the results of this study are demonstrated in table 6 and are suitably generic; they can be applied across all areas of healthcare as they are not limited to orthopaedic rehabilitation. Illustrations with sensitising questions (online supplemental material 4 = Situation of care, online supplemental material 5 = Expectations of care, online supplemental material 6 = Capacity to allocate resources to care, online supplemental material 7 = Demands of care) are presented within the online supplemental materials.

### Results in context

Burden of treatment theory[24] and the cumulative complexity model[19] both focus on the relationship between the workload demands on the patient with the patients capacity to do the work. Our previous research[15] hypothesised that the work of being a patient influences

**Table 6** Practical questions to support formation of preference

| Theme | Factor | Description | Practical questions to support identification of preference for patients | Practical questions for clinicians to ask patients to support identification of preference |
|-------|--------|-------------|--------------------------------------------------------------------------|----------------------------------------------------------------------------------------------|
| Situation of care | Clinical status | The healthcare complaint the patient experiences, its stability, reversibility and its impact on the patient in conjunction with other complaints. | ▶ Does my problem require me to be seen in person?<br>▶ Would having a virtual appointment make things easier for me? | ▶ Does your problem require you to be seen in person?<br>▶ Would having a virtual appointment make things easier for you? |
| | Treatment requirements | The treatment and management of the complaint that is required. The restrictions imposed on the patient. | ▶ Can the treatment I need be delivered virtually? | ▶ Do you think the treatment you need can be delivered virtually? |
| | Care pathway | The availability of healthcare to the patient | ▶ What do I need from my clinician to support me with a face-to-face or a virtual appointment? | ▶ What can I do to support you with a face-to-face or a virtual appointment? |
| Expectations of care | Desire for contact | Whether the patient / healthcare professional believes the F2F is more of a capable method of care delivery than VC. | ▶ Do I think my issue can be best managed by a face-to-face or a virtual appointment?<br>▶ Does my healthcare professional think my issue can be best managed by a face-to-face or a virtual appointment? | ▶ Do you think your issue could be best managed by a face-to-face or a virtual appointment?<br>▶ Do you believe I think your issue could be best managed by a face-to-face or a virtual appointment? |
| | Psychological status | The psychological status of the patient and the impact of this on care across different delivery formats. | ▶ How would a virtual appointment affect me?<br>▶ Am I comfortable seeing myself on a screen? | ▶ How would a virtual appointment affect you?<br>▶ Would you be comfortable seeing yourself on a screen? |
| | Previous care | Experience of previous care | ▶ Could my previous treatment have been managed successfully virtually? | ▶ Do you think your previous treatment could been managed successfully virtually? |
| | Perceived requirements | The negotiated requirements of the session | ▶ How can my problem be managed best?<br>▶ Can my problem be managed by a face-to-face or virtual appointment? | ▶ How can your problem be managed best?<br>▶ Can your problem be managed by a face-to-face or virtual appointment? |
| Demands of care | Care requirements | The requirements of care | ▶ What do I need to during my rehab?<br>▶ Can I achieve this? | ▶ What does your care require of you?<br>▶ Can you achieve this? |
| | Social demands | The competing life demands that can interfere with healthcare. | ▶ What other things do I need to do that might get in the way of a F2F or VC? | ▶ What other things do I need to do that might get in the way of a F2F or VC? |
| | Consequences of choice | The impact of choice | ▶ What do I need to do if I choose a face-to-face or a virtual appointment? | ▶ What do you need to do if you choose a face-to-face or a virtual appointment? |

**Table 6** Continued

| Theme | Factor | Description | Practical questions to support identification of preference for patients | Practical questions for clinicians to ask patients to support identification of preference |
|---|---|---|---|---|
| Capacity to allocate resources to care | Financial | The ability to free up financial resources | ► What would the financial impact be for me if I choose a face-to-face or a virtual appointment? | ► What would the financial impact be for you if you choose a face-to-face or a virtual appointment?? |
| | Infrastructure | Access to material and informational resources | ► Do I have access to what I need to have a face-to-face or a virtual appointment?<br>► Do I understand how to use what is needed for a virtual appointment? | ► Do you have access to what you need to have a face-to-face or a virtual appointment??<br>► Do you understand how to use what is needed for a virtual appointment? |
| | Social capacity | Support available through social network | ► Do I have anyone who could support me with a face-to-face or a virtual appointment? | ► Do you have anyone who could support you with a face-to-face or a virtual appointment? |
| | Healthcare system | Sources of healthcare capacity | ► How can my healthcare professionals support me to access my care with either a face-to-face or a virtual appointment? | ► How can we support you to access your care with either a face-to-face or a virtual appointment? |

F2F, face-to-face consultation; VC, virtual consultation.

preference; patients may prefer the least burdensome option when giving the choice between a F2F and VC.

This current paper extends our previous model of patient preferences adding in: the situation of care, patient's expectations of care and patients ability to allocate resources to care (see figure 2). Some patients find the process of F2F attendance burdensome. Despite this, some of these patients preferred to receive hands on manipulation. Some patients were prepared to tolerate burden as part of a process that offered them F2F care they believed was superior to a VC. In addition, some patients perceived the consequences of choosing a F2F (or VC) would significantly impact on their overall experience of care, both positive or negative. Additionally, factors such as confidentiality in VC and trustworthiness[27] may influence expectations of care. The model within this paper clearly demonstrates additional factors relating to BoT are likely to influence their preference. The option that best meets patients' expectations of care influences preferences.

Some patients discussed the situational nature of their problem and how their preferences may have been different under different circumstances. This is in accord with our qualitative study of acceptability for rehabilitation consultations.[14] Greenhalgh et al[10] found that videoconferencing using Skype was useful to access hard to reach patients and that avoiding long journeys to access care was beneficial. Not travelling can reduce healthcare costs[28] and the need for family to accompany patients on their journey.[14] Patients without the support of their families in our study found this to be beneficial. Kaambwa et al[13] found that patients had strong preferences for VCs when their clinic was between 15 to 100 km away and when their use reduced costs. The dynamics between the situation and the patient's capacity for care create a unique state of affairs for each patient at the time of being offered the choice between consultations. These factors directly influence the patients burden and expectations of care. Consideration of these factors, and identification of the option with the most utility to the patient, will influence preferences.

This study is separated from many others (eg, in primary care[29] and psychiatry[30] studies) because orthopaedic rehabilitation often requires 'hands on' care which is not possible virtually. The lack of touch over VC can inhibit patients experience of receiving care, particularly when they desire it.[31] Patients in the PhysioDirect study of telephone consultations still wanted to have 'proper' F2F physio.[32] VC has been seen as 'impersonal'[33] and can reduce emotional bonding between the patient and healthcare professional.[31]

A common theme in our data was the negative psychological impact some patients felt seeing themselves through a screen. This was in accord with a patient in the

Jansen-Kosterink study[33] who reported: 'I cannot imagine seeing myself on video, I already have trouble seeing myself in a picture'. Some patients for whom this was not a problem, however, found that being in their own environment and avoiding travel made them feel more relaxed[10] which could in itself improve patient–healthcare professional relationships. If offered the choice of a F2F or VC, patients need to give consideration to the alternatives; the actions, the state of affairs and the consequences of choosing each alternative. The present research does not suggest how *much* the highlighted factors influences preferences or compete and compete with each other. This study will inform the design of a Discrete Choice Experiment, a deductive investigation to quantitatively measure how each factor influences preferences for patients in a pragmatic real-world scenario. A thorough understanding of the effect and influence of preferences will enable patient-centred service design.

However, the results of this study should be interpreted in light of their limitations. It was conducted at a single centre and may not translate to other clinical areas. To overcome this, variation across participants was sought and attention focused towards more general factors to allow for transportability to other clinical settings. The lead researcher (AWG) is a healthcare professional within the centre which could have led to bias results through local familiarity. To limit this, patients who had a previous existing relationship with AWG were excluded from the study as per the exclusion criteria. It was not possible, however, to exclude clinical staff, most of whom were known to AWG. This was taken into account in the data analysis through a process of defamiliarisation; attributions for each data point were orientated into a taxonomy to facilitate model development. Raw interview data was used to illustrate the model.

### Potential impact of COVID-19 pandemic on the future of virtual consultations

The empirical data collection for this research was conducted prior to the COVID-19 pandemic. The COVID-19 pandemic has accelerated the introduction of VC across healthcare. The rapid implementation of VC[9] may shape the future of this work in a way that was not previously anticipated. The COVID-19 'situation' has influenced an increased uptake of VC in practice. While this research did not formally collect data regarding previous experience of VC (even in a different setting), future research should explore patient and clinician experience of using VC for healthcare consultations. Further research evaluating the use of VC during the COVID-19 pandemic will support future service redesign.

### CONCLUSIONS

We conducted 44 qualitative interviews to gain a thorough understanding of the mechanisms that influence patient preference. Multiple factors were identified: the situation of care (the ways that patients understand and explain their clinical status, their treatment requirements and the care pathway), the expectations of care (influenced by a patients desire for contact, psychological status, previous care and perceived requirements), the demands of care (of each patients respective social situation and the consequences of choice) and the capacity to allocate resources to care (the patient's ability to allocate resources to care; these include financial, infrastructural, social and healthcare resources). Factors may combine or compete with each other to influence preference. The patient's situation is dynamic and therefore preferences must also be dynamic. The formation of preference is cognitively demanding and sensitising questions may support patients to identify their preferred consultation format. This research illuminates the factors that appear to influence preference for patients. This is important for healthcare professionals; an understanding of preferences is essential to support the design of patient care pathways incorporating virtual consultations. The dynamic model presented here can be used to inform quantitative studies such as discrete choice experiments, and could act as a programme theory to inform future trials.

**Acknowledgements** The authors thank Rachel Dalton and members of the CONNECT Project patient and public involvement steering group for their invaluable contributions to the overall study design of the CONNECT Project and obtaining funding for the PhD Fellowship. The authors thank Kate Forrester and Geoff Buckley for their assistance with the development of the patient facing materials. The authors also thank John Doyle, Anju Jaggi, Iva Hauptmannova and colleagues within the Therapies Directorate and Research and Innovation Centre at the Royal National Orthopaedic Hospital for their ongoing support. The authors are grateful to the 22 patients and 22 members of staff who participated in this study.

**Contributors** AWG wrote the paper and conceived the project with CRM, JJ and MS. CRM guided qualitative data collection. AWG conducted all the interviews. CRM assisted with data analysis, and with AWG developed the model. CRM, JJ and MS edited and critically revised the paper. All authors have read and approved the manuscript. AWG is the guarantor of the manuscript.

**Funding** Anthony Gilbert is funded by a National Institute for Health Research (NIHR), Clinical Doctoral Research Fellowship for this research project (ICA-CDRF-2017-03-025). Anthony Gilbert and Carl May are supported by the NIHR ARC North Thames. The views expressed in this publication are those of the author(s) and not necessarily those of the NIHR or the Department of Health and Social Care.

**Competing interests** None declared.

**Patient consent for publication** Not required.

**Ethics approval** Ethical approval was received for this study (approval received on 4 December 2018 from the South Central-Oxford C Research Ethics Committee (IRAS ID: 255172, REC Reference 18/SC/0663)). All participants were provided with a participant information sheet and given at least 24 hours to consider the information and ask questions before being recruited into the study. All participants provided informed, written consent prior to enrolment.

**Provenance and peer review** Not commissioned; externally peer reviewed.

**Data availability statement** Data are available upon reasonable request. Data are available upon reasonable request from the corresponding author.

**ORCID iDs**

Anthony W Gilbert http://orcid.org/0000-0003-2526-8057
Maria Stokes http://orcid.org/0000-0002-4204-0890
Carl R May http://orcid.org/0000-0002-0451-2690

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
