## [Reviewer comments · BMJ Open]

ARTICLE DETAILS

TITLE (PROVISIONAL)	Factors that influence patient preferences for virtual consultations in an orthopaedic rehabilitation setting: a qualitative study
AUTHORS	Gilbert, Anthony; Jones, Jeremy; Stokes, Maria; May, Carl

VERSION 1 – REVIEW

REVIEWER	Maria Larsson University of Gothenburg, Institution of Neuroscience and Physiology, Dept of Health and Rehabilitation, Sweden
REVIEW RETURNED	28-Jun-2020

GENERAL COMMENTS	Thank you for the opportunity to review this manuscript. It has an interesting topic. I find preferences for treatment, and treatment delivery to be of great interest and highly relevant, especially these days where we have changed our treatment delivery models to a great extent to use virtual consultations. I believed there is material with great potential to become a good paper. I do not mean to be rude in any way saying this. My main concern with the paper in its' present shape is that I find the material mainly sorted and abstracted but not very thoroughly analysed in regard to the research question as it is described "...aims to identify, characterise, and explain factors that influence patient preferences for VCs in an orthopaedic rehabilitation setting". I find the themes to be abstracted to a level that makes them have such general and universal headings so it can be applied to anything and everything. The subthemes also have very general headings and, this makes them loose meaning. I read through the text wondering what and how over and over again. And thinking: how to use this in my clinical setting or for further research? Reading the descriptions of the subthemes gives more information as it is closer to data, but it is hard to identify its' specificity to the subtheme and theme. Even though it is (sub)themes (and not categories) I find them maybe too often to intervene with each other as for "care factors" and "clinical factors" or as for the question about travel which is included both in "care pathways" as in "psychological status", "perceived requirements", "consequence of choice", "financial", "social". And how are "Social factors" and "social" different concepts from each other even described in different themes? With so universal headings it loses meaning to me in this particular area of interest. This means that I read over and over again but still do not know how to use the information, so in its current shape it is not very informative. Even though the title says it is an abductive approach to the analysis I do expect to get more specific answers to the what factors and how they influence patient preferences to use communication technology. Abstract
--

	I would really like to have more explicit results and conclusion. There is a statement that a conceptual model has been developed, but the abstract does not give much information about what the preferences are? What in the factors are influencing preferences? That a current situation or expectations influence preferences are not particularly novel or give much information of what and how. The key factors would be much more interesting if they included more substance. Background Page 2 line 38 prefer a people's first language ie patients with stroke. Are there no other studies on patient preferences in for example physiotherapy? Capacity as well as demand are mentioned in the background and also introduced as parts of BoT. Later on, these factors arised from data. Is there a risk that the study did not have an abductive but a deductive approach? Methods I am not sure what you mean by maximum variation sample? There is no reference to this. Either describe in more detail or use a reference. How come health professionals were interviewed, as the interest was what influence patient preference? Were the two pilot interviews included in the analysis? How was, or was the interview schedule adjusted after the pilot interviews? How many showed interest to participate in the study? Were there any selection among those interested? What was the rational for the decision of 22+22 participants? The data analysis is very briefly described. Uploaded into what Nvivo version? Who did what during the analysing process? Where all authors involved in every step of the process? How where disparate opinions solved? Open coding was used for data analysis, as in Grounded Theory? Please provide reference. There are no references at all to the analysing process. The abstract states that an abductive analysis informed by Burden of Treatment Theory was used. I cannot see this in the methods section? Results I would like to see a more detailed description of the participants, except for age, sex, and profession. Preferably, musculoskeletal conditions and for the professionals also for experience from the setting. Is there an overarching theme? I do not understand the subheading "interpretation of results"? Do you mean as examples on how the theme has been constructed? I have already given my main comments above regarding the results; There is always a situation to handle, what is specific about the situation for the preferences to use or not use VC? There is a mapping but I find no "meaning" to it. Nor the themes or the subthemes give much more than a sorting title. After reading the whole results I am informed about four factors but I still wonder, and? I think you need to help and guide the reader to the interpretation (and maybe to use in conclusion) of your results. Now you have to go all the way to read all the text to understand what it means. The model is interesting and should be explored and described to a greater extent in the results. Could you have come to the same model purely on theoretical studies, with no empirical data? Or with a deductive approach?
--	--

	There is a model/figure in the supplemental files, in Part 2 – Results of Phase 1 – what is this? A model? Part of the interview guide? Or part of results? Here again you have expectations, you have environment which is very close to situation. You have time, logistics, resources and so forth which is capacity. Discussion Summary of results – I think this is part of results, not discussion. I lack a more thorough discussion in regard to the concepts of trustworthiness. Conclusion The conclusion does not need to repeat the results and could be more concrete in what understanding have we received from this study and how to use it. I find it to be very general. Minor comments Page 9, line 21 edit “pofessionals”
--	---

REVIEWER	Emma Phelps University of Oxford, UK
REVIEW RETURNED	25-Aug-2020

GENERAL COMMENTS	Thank you for the opportunity to review this interesting manuscript about factors that influence patient preferences for video consultation in a rehabilitation setting. This is a timely topic that could be beneficial to clinical practice. Below are a few suggestions which I believe may strengthen the paper. The manuscript would also benefit from a thorough proof read and several sentences which are unnecessarily complicated could be simplified. Abstract  1. Please clarify the purpose of the clinician interviews. In the strengths and limitations you state “this is the first qualitative investigation of patient and clinician preferences for video consultation” but in the objective and throughout the rest of the text only patient preferences are mentioned. Were you looking at clinician preferences as well or patient preferences from the perspective of patients and clinicians? 2. Full stop missing at the end of the objective. 3. The second Twenty-two in line 20 should not be capitalised. 4. In the conclusion of the abstract you state “This study has identified key factors that appear to influence patient preference for video-conferenced consultations in orthopaedic rehabilitation. A robust conceptual model of these factors has been developed highlighting how they combine and compete”. This is in contrast to your discussion where you state your research does not suggest how these factors compete with each other. Which is correct? Additionally are you able to say your conceptual model robust at this stage? Methods  1. Please clarify the purpose of the clinician interviews. 2. “All transcripts were emailed or posted to participants upon receipt to give them the option to verify the data or to make any adjustments” – Did any participants make adjustments and if so please explain how this may have affected your results? 3. Who conducted the interviews and analysed the data? In the discussion you state “The lead researcher (AG) is a healthcare professional within the centre” - Was there an existing relationship between the interviewer and interviewees (patients and clinicians)
---

	and if so how may this have influenced the data you collected? If you interviewed your own patients please discuss the ethical implications associated with doing this. 4. You have included a topic guide for interviews with patients. Do you have a clinician interview topic guide? 5. Did any of your participants have experience of using video consultations (even in another clinical setting)? Patients and clinicians with experience of VC may have responded differently to those without experience. I would suggest that future research on this topic should explore patient and clinicians experience of using VC for orthopaedic rehabilitation consultations. 6. Providing a reference for your method of analysis might be helpful to readers. Results 1. I am not sure it is necessary to include the age of the clinicians. Perhaps years of experience would be a more helpful demographic? 2. Page 8 Line 41 – should this be way rather than sway. 3. At times in your results it is not clear if you are reporting the views of participants or your own views/knowledge. For example in Care Factors (Theme 3) there is nothing to relate your text to the data or participants. 4. What are social requirements – are they the same as social demands? If so, being consistent would be helpful to the reader (Social Factors – Theme 3). 5. “The use of equipment requires a set of skills that was not familiar to some patients in this study, including rehabilitation equipment and technology for VC” (Page 10) – This sentence is unnecessarily complicated - could this be written more clearly? 6. The description of Consequences of Choice (in Theme 3) “The things patients need to do as a direct consequence of the choice made” doesn’t match the text or quotes you provided- some of these consequences are not things that patients need to “do”. 7. (Page 12, Line 16) Healthcare professionals were aware of these to these financial challenges faces by patients – remove “to these”. 8. (Page 12, Line 22) There was variation in the types of technology utilized and it was not overtly obvious that one particular type influenced preference although some devices with larger screens were thought to be more beneficial and influence expectations – this sentence is confusing, could it be simplified. 9. (Page 12, Line 40) The situation that the patient is orientated to can provide capacity. – This sentence does not make sense to me. Discussion 1. The model (figure 1) explaining the relationship between the factors is complicated. The description is also complicated and repetitive. I think you could explain this much more simply. 2. (Page 14) The capacity and available resources of a patient may influence their expectations of care. – You have defined available resources to be part of capacity so available resources is not needed in this sentence. 3. (Page 15) For some patients the physical attendance at a clinic was burdensome, yet they preferred to attend due for a particular experience of care; for example to receive hands on manipulation – Could this be simplified? 4. (Page 16) Flexible options for patients and healthcare professionals to were provided to participate, both over phone and
--	--

	Skype as well as F2F. This sentence doesn't make sense. Could it be rephrased? 5. (Page 16) Multiple factors were identified: The situation- 'The' should not be capitalised. 6. I would include a clear sentence highlighting the implications of your findings for clinical practice.
--	--

VERSION 1 – AUTHOR RESPONSE

Response to reviewers (bmjopen-2020-041038)

Reviewer Comment	Response
Reviewer: 1	
Thank you for the opportunity to review this manuscript. It has an interesting topic. I find preferences for treatment, and treatment delivery to be of great interest and highly relevant, especially these days where we have changed our treatment delivery models to a great extent to use virtual consultations. I believed there is material with great potential to become a good paper. I do not mean to be rude in any way saying this. My main concern with the paper in its' present shape is that I find the material mainly sorted and abstracted but not very thoroughly analysed in regard to the research question as it is described "...aims to identify, characterise, and explain factors that influence patient preferences for VCs in an orthopaedic rehabilitation setting".	We have attempted to lead the reader through this process. We have identified 4 factors and have characterised these within the results. We have offered an explanation as to how these factors interact within the discussion section of the paper and have offered an explanatory model to illustrate this.
I find the themes to be abstracted to a level that makes them have such general and universal headings so it can be applied to anything and everything.	This was the purpose. We did not want this paper to be specific to orthopaedic rehabilitation consultations. With this in mind we reduced concepts to their simplest form in order to make the explanations transportable to other areas.
The subthemes also have very general headings and, this makes them loose meaning. I read through the text wondering what and how over and over again.	Please see the comment in the box immediately above.
And thinking: how to use this in my clinical setting or for further research?	We have outlined how these results could be applied in table 3. In summary, this table offers practical questions to support patients to form preferences. Table 3 is located in page 17. For instance, for the theme 'situation' factor 'clinical status' the questions we have suggested are:

	 • Does your complaint need to be seen? • Would having a VC make things easier for you? For 'treatment requirements'  • Can the treatment you require be delivered virtually? And for 'care pathway'  • What can we do to support you with a F2F or VC? And so on.
Reading the descriptions of the subthemes gives more information as it is closer to data, but it is hard to identify its' specificity to the subtheme and theme. Even though it is (sub)themes (and not categories) I find them maybe too often to intervene with each other as for "care factors" and "clinical factors" or as for the question about travel which is included both in "care pathways" as in "psychological status", "perceived requirements", "consequence of choice", "financial", "social".	These factors do overlap and are not distinct. In response to your point about travel – the 'care pathway' includes the length of the appointment, number of appointments and regularity of these and the time of day of the appointments. Some patients found it difficult to travel in rush hour (particularly early mornings) for example. Travel is relevant in 'psychological status' – some patients were anxious about the travel because of the amount of pain they were in. We chose not to focus on 'travel' as a separate entity because it affects many aspects relating to preference, as we have demonstrated. We wanted to avoid being too simplistic with this.
And how are "Social factors" and "social" different concepts from each other even described in different themes? With so universal headings it loses meaning to me in this particular area of interest. This means that I read over and over again but still do not know how to use the information, so in its current shape it is not very informative.	This has been a very helpful point. We have relabelled 'social factors' to 'social demands' (p12). We feel, based on your comments, the article is stronger with 'table 3' (page 17) which goes some way to answering the "so what" question of the research as it offers a practical way to use the findings in clinical practice to identify and facilitate discussions surrounding patient preference for virtual consultations. Please refer to table 3 which offers a practical way to use these data to support the formation of preferences.
Even though the title says it is an abductive approach to the analysis I do expect to get more specific answers to the what factors and how they influence patient preferences to use communication technology.	We have identified and characterised the factors in the results. These factors were identified through seeking variation within data and abductive explanations have been offered in accordance with the methodology and method of abductive analysis. We have rewritten

	elements of the discussion in the hope that this more clearly sets out how the factors may influence preference.
Abstract I would really like to have more explicit results and conclusion. There is a statement that a conceptual model has been developed, but the abstract does not give much information about what the preferences are? What in the factors are influencing preferences? That a current situation or expectations influence preferences are not particularly novel or give much information of what and how. The key factors would be much more interesting if they included more substance.	We have spelled these out. Hopefully these read as having more substance. (abstract section)
Background Page 2 line 38 prefer a people's first language ie patients with stroke.	This has been amended as suggested
Are there no other studies on patient preferences in for example physiotherapy?	Not to our knowledge in relation to telemedicine.
Capacity as well as demand are mentioned in the background and also introduced as parts of BoT. Later on, these factors arised from data. Is there a risk that the study did not have an abductive but a deductive approach?	Burden of treatment underpins this work: the interview schedule was underpinned by burden of treatment theory and the results of our previous work. The process of abduction took place during analysis to identify additional factors to those identified during our work from phase 1. Within capacity and demand in the results we sought variation and explanation which is a process of abduction. The process followed has been illustrated using a visual diagram within the methods section.
Methods I am not sure what you mean by maximum variation sample? There is no reference to this. Either describe in more detail or use a reference.	
How come health professionals were interviewed, as the interest was what influence patient preference?	We have made this clear in page 4 in the 'participants' para: 'Clinicians were selected to be interviewed to provide their perspectives on patient preference and as patient preferences are moderated by

	the possibilities and preferences of organisations and staff.'
Were the two pilot interviews included in the analysis?	No changes were made to the interview schedule after the pilot interviews – these data were included in the analysis after consultation between authors.
How was, or was the interview schedule adjusted after the pilot interviews? How many showed interest to participate in the study?	No changes were made to the interview schedule following the pilot interviews – we have stated this at the top of the results section on page 7. We did not collect this data at the time, as participants were alerted to the study via a pop-up banner there may have been patients who did not show interest. Ultimately, the 22 patients who participated showed interest in the study.
Were there any selection among those interested?	Apologies, we do not know what you mean by this comment.
What was the rationale for the decision of 22+22 participants?	We opted for 20 patients (mixture of ages) and 20 healthcare professionals (mixture of professions) to allow as wide a variation as possible within the feasibility of a PhD project. We were not looking for data saturation, more insights generated from the data and from previous experience it was agreed this number was likely to be sufficient to generate insights.
The data analysis is very briefly described.	This section has been re-written extensively in accordance with comments from both reviewers. We trust the section now has sufficient detail. (page 5, data analysis)
Uploaded into what Nvivo version?	It was uploaded into NVIVO version 12. we have added this within the paper. (page 5, data analysis)
Who did what during the analysing process?	This section has been rewritten, to clarify the major pieces of work within the analysis:  • Coding was undertaken by AG and CM • Attributions were assigned by AG, CM and reviewed by JJ. • The model was mapped out by AG and reviewed by CM, JJ and MS. (page 5, data analysis)
Were all authors involved in every step of the process?	No, please see response to 'who did what during the analysis process'. A description of the

	author roles is within page 5 – data analysis section.
How where disparate opinions solved?	Via discussion. JJ was assigned this role in case of disparate opinions. This was not needed.
Open coding was used for data analysis, as in Grounded Theory? Please provide reference.	Open coding was used to identify these data in accordance with Charles S Pierce’s semiotics. We have provided a reference to Tavory and Timmerman’s book on abductive analysis which sets out the epistemological underpinnings and methods for this.
There are no references at all to the analysing process.	We have added ‘Data analysis followed the principles of abduction as set out by Tavory and Timmermans’ and referenced the text in which the analysis method was founded upon. (page 5, line 2, data analysis para)
The abstract states that an abductive analysis informed by Burden of Treatment Theory was used. I cannot see this in the methods section?	We have removed burden of treatment theory from the abstract. We used open coding so as not to limit to constructs of Burden of Treatment.
Results I would like to see a more detailed description of the participants, except for age, sex, and profession. Preferably, musculoskeletal conditions and for the professionals also for experience from the setting.	Unfortunately, we do not have access to these data but we do agree it would have been helpful to have their number of years of experience. We did not collect musculoskeletal conditions – we decided to do this because the purpose of this research was to generate transportable insights and not be condition specific.
Is there an overarching theme?	No, there were counter arguments in favour of and against VC. We have illustrated the key themes identified within the results.
I do not understand the subheading “interpretation of results”? Do you mean as examples on how the theme has been constructed?	We have changed to ‘Characterisation of themes’.
I have already given my main comments above regarding the results;	
There is always a situation to handle, what is specific about the situation for the preferences to use or not use VC?	We have tried to make this clearer within the text and simplified the language used, in accordance with reviewer 2’s comments. Ultimately, the situation inform expectations, demands and capacity as we have stated within the text. We have attempted to illustrate this within the results and discussion section. We agree there is always a situation to handle.

There is a mapping but I find no “meaning” to it. Nor the themes or the subthemes give much more than a sorting title.	We have characterised the themes and subthemes. In addition, Table 3 offers a way to clinically apply these results in practice.
After reading the whole results I am informed about four factors but I still wonder, and?	We have developed a table of potential questions to support clinicians to identify patient preferences through subjective questioning. Table 3 is on page 17.
I think you need to help and guide the reader to the interpretation (and maybe to use in conclusion) of your results. Now you have to go all the way to read all the text to understand what it means.	Please see the comments above and refer to table 3 on page 17. Here is a table of practical questions to support the formation of preferences. These questions have been generated based on the results of this study.
The model is interesting and should be explored and described to a greater extent in the results.	The model is an interpretation and explanation of the results.
Could you have come to the same model purely on theoretical studies, with no empirical data? Or with a deductive approach?	It is a possibility that a similar model could be devised through either theoretical studies or a deductive approach. It is important to note, however, that a strength of this paper is that the theoretical insights and explanations generated from this paper are developed from empirical data and are likely to be meaningful to clinical practice. We have added a statement to that effect in ‘strengths and limitations of this study’. (page 2)
There is a model/figure in the supplemental files, in Part 2 – Results of Phase 1 – what is this? A model? Part of the interview guide? Or part of results? Here again you have expectations, you have environment which is very close to situation. You have time, logistics, resources and so forth which is capacity.	These are the results of the qualitative systematic review we previously conducted. As stated within the methods section, the interview guide was developed using the results of Phase 1. AG described these results to participants and demonstrated the model to participants to stimulate discussion. This forms part of the interview guide. We trust this response answers the reviewer’s question.
Discussion Summary of results – I think this is part of results, not discussion.	We have removed the heading ‘summary of results’. This section intends to frame the results section – it has been reworded slightly to recap the fact that factors have been identified and characterised, and that the following discussion section now offers an explanatory model to demonstrate the interplay between factors. (page 15-16)
I lack a more thorough discussion in regard to the concepts of trustworthiness.	Thank you for this suggestion. Although this is a small aspect of what we are investigating, we agree this is very important.

	Within the discussion we have added the following sentence 'Additionally, factors such as confidentiality in VC and trustworthiness may influence expectations of care' and referenced Mark Hall's work. (page 17, penultimate para)
Conclusion The conclusion does not need to repeat the results and could be more concrete in what understanding have we received from this study and how to use it. I find it to be very general.	Having discussed this amongst the authors, we have made a small amendment to the conclusion. We would like to add that the purpose of this paper was not to provide a descriptive account of preferences but to identify what these factors are (this is in the conclusion) and characterise them (we have added a description for each factor). The explanatory element of this paper is situated in the discussion and can be briefly summarised by the sentence 'Factors may combine or compete with each other to influence preference.' (page 19) We have made a change in the penultimate para of the conclusion 'This research illuminates the factors that appear to influence preference for patients. This is important for healthcare professionals; an understanding of preferences is essential to support the design of patient care pathways incorporating videoconferencing for consultations' which we trust sets out how the results of this study can be applied. (page x, para x, lines x-x)
Minor comments Page 9, line 21 edit "pofessionals"	We have edited, thank you.
Reviewer: 2	
Thank you for the opportunity to review this interesting manuscript about factors that influence patient preferences for video consultation in a rehabilitation setting. This is a timely topic that could be beneficial to clinical practice. Below are a few suggestions which I believe may strengthen the paper. The manuscript would also benefit from a thorough proof read and several sentences which are unnecessarily complicated could be simplified.	We have been through the paper and have made amendments throughout to try and simplify the work. Thank you for your comments, these changes have strengthened the paper and we are very grateful.

Abstract 1. Please clarify the purpose of the clinician interviews. In the strengths and limitations you state “this is the first qualitative investigation of patient and clinician preferences for video consultation” but in the objective and throughout the rest of the text only patient preferences are mentioned. Were you looking at clinician preferences as well or patient preferences from the perspective of patients and clinicians?	The objective within the abstract now reads as ‘To identify characterize and explain factors that influence patient preferences, from the perspective of patients and clinicians, for virtual consultations in an orthopaedic rehabilitation setting.’ as we were looking at patient preferences from the perspective of patients and clinicians. (abstract) We have added the following sentence within the manuscript which we believe justifies the importance of clinician interviews in relation to patient preferences. ‘Clinicians were selected to be interviewed to provide their perspectives on patient preference and as patient preferences are moderated by the possibilities and preferences of organisations and staff’ (page 4, ‘participants’ para) The strengths and limitations statement now reads as ‘This is the first qualitative investigation of patient preferences for video consultation in a tertiary orthopaedic setting.’
2. Full stop missing at the end of the objective.	We have added a full stop.
3. The second Twenty-two in line 20 should not be capitalised.	We have changed this to lower case.
4. In the conclusion of the abstract you state “This study has identified key factors that appear to influence patient preference for video-conferenced consultations in orthopaedic rehabilitation. A robust conceptual model of these factors has been developed highlighting how they combine and compete”. This is in contrast to your discussion where you state your research does not suggest how these factors compete with each other. Which is correct?	The discussion section says that the research does not suggest how much these factors influence preference. We have reworded in a way that hopefully makes it clearer for the reader: ‘The present research does not suggest how much the highlighted factors influences preferences or combine and compete with each other’ (page 19, para 4)
Additionally are you able to say your conceptual model robust at this stage?	We have removed the word robust.

Methods 1. Please clarify the purpose of the clinician interviews.	We have added the following sentence which we believe justifies the importance of clinician interviews in relation to patient preferences. 'Clinicians were selected to be interviewed to provide their perspectives on patient preference and as patient preferences are moderated by the possibilities and preferences of organisations and staff' (page 4, 'participants' para)
2. "All transcripts were emailed or posted to participants upon receipt to give them the option to verify the data or to make any adjustments" – Did any participants make adjustments and if so please explain how this may have affected your results?	We have added the following sentence into the results section: "No participants returned their transcripts and therefore no amendments were made." (page 7, para 1)
3. Who conducted the interviews and analysed the data? In the discussion you state "The lead researcher (AG) is a healthcare professional within the centre" - Was there an existing relationship between the interviewer and interviewees (patients and clinicians) and if so how may this have influenced the data you collected? If you interviewed your own patients please discuss the ethical implications associated with doing this.	We have added the following into data collection: 'Interviews were conducted by AG and were to last around 60 minutes with the option to extend or shorten as required.' (page 4, final para) We have added the following sentences in data analysis (which has largely been rewritten) 'Coding was undertaken by AG.' And 'Attributions were assigned to codes following discussion between AG & CRM.' And 'Attributions were subsequently discussed between AG and JJ to ensure they made sense and were accurate representations of these data. No changes were required to attributions at this stage.' 'The theoretical model was discussed between all authors to verify its content.' We have added the following into the exclusion criteria which was within the protocol for the study but omitted from the paper 'Patients

	currently or previously treated by AG'. (table 1, page 6) We have rewritten the section within the Discussion, which now reads as: 'The lead researcher (AG) is a healthcare professional within the centre which could have led to bias results through local familiarity. To limit this, patients who had a previous existing relationship with AG were excluded from the study as per the exclusion criteria. It was not possible, however, to exclude clinical staff, most of whom were known to AG. This was taken into account in the data analysis through a process of defamiliarization; attributions for each data point were orientated into a taxonomy to facilitate model development.' (page 18, penultimate para)
4. You have included a topic guide for interviews with patients. Do you have a clinician interview topic guide?	Clinicians were asked general questions in relation to the results from phase 1 and their supposed preferences of patients. We have attached the clinician version of the interview guide. (supplementary material 2)
5. Did any of your participants have experience of using video consultations (even in another clinical setting)? Patients and clinicians with experience of VC may have responded differently to those without experience. I would suggest that future research on this topic should explore patient and clinicians experience of using VC for orthopaedic rehabilitation consultations.	We do not know the answer to this because we did not ask this question of participants. We agree with the reviewer that future research should include experience and we have added the following in to the 'Potential impact of Covid-19 pandemic on the future of videoconferencing' part of the discussion section: 'Whilst this research did not formally collect data regarding previous experience of video consultations (even in a different setting), future research should explore patient and clinician experience of using VC for healthcare consultations.' (page 19 para 1)
6. Providing a reference for your method of analysis might be helpful to readers.	We have added 'Data analysis followed the principles of abduction as set out by Tavory and Timmermans' and referenced the text in which the analysis method was founded upon. (page 5, line 2, data analysis para)

Results 1. I am not sure it is necessary to include the age of the clinicians. Perhaps years of experience would be a more helpful demographic?	We have removed age of clinicians. Unfortunately, we do not have access to these data but we do agree it would have been helpful to have their number of years of experience.
2. Page 8 Line 41 – should this be way rather than sway.	Many thanks for pointing this out, we have amended this.
3. At times in your results it is not clear if you are reporting the views of participants or your own views/knowledge. For example in Care Factors (Theme 3) there is nothing to relate your text to the data or participants.	Hopefully elsewhere in the comments (and the inclusion of the clinician interview schedule) we have now made it clear that we were asking questions in relation to patient preferences and what is being reported are patient preferences. We have changed the heading 'interpretation of results' to 'interview data' We have spelt out that the data reports both healthcare professionals and patients' responses: 'The resulting data can be seen in Table 2a-2d and reports data for patients and healthcare professionals.' Rather than call this section 'interpretation of results' we have referred to this as 'Characterisation of themes'. This is in relation to the objectives of the paper: to 'to identify, characterise, and explain factors that influence patient preferences for VCs in an orthopaedic rehabilitation setting.' The 'identification' and 'characterisation' sits within the results section (page 6 - 14) and the 'explain' part of this sits within the discussion section (Figure 1 & within the discussion section page 15-16).
4. What are social requirements – are they the same as social demands? If so, being consistent would be helpful to the reader (Social Factors – Theme 3).	Thanks for this comment and we agree it may be a source of confusion. We have reworded these to 'social capacity' and 'social demands'. We have changed 'social requirements' to 'social demands'. (p11)
5. "The use of equipment requires a set of skills that was not familiar to some patients in this study, including rehabilitation equipment and technology for VC" (Page 10) – This	We have reworded this to: 'The use of virtual consultation equipment may require a new skill set. Patients might also need

sentence is unnecessarily complicated - could this be written more clearly?	to obtain rehabilitation equipment and technology for VC.’ (page 11. Consequences of choice para)
6. The description of Consequences of Choice (in Theme 3) “The things patients need to do as a direct consequence of the choice made” doesn’t match the text or quotes you provided- some of these consequences are not things that patients need to “do”.	We have changed this description to ‘The impact of choice’. (table 2c, page 12)
7. (Page 12, Line 16) Healthcare professionals were aware of these to these financial challenges faces by patients – remove “to these”.	Many thanks, we have removed.
8. (Page 12, Line 22) There was variation in the types of technology utilized and it was not overtly obvious that one particular type influenced preference although some devices with larger screens were thought to be more beneficial and influence expectations – this sentence is confusing, could it be simplified.	We have reworded to: ‘Variations of hardware and software exist. There did not appear to be any relationship with type of hardware and software combination and preference. Some devices with larger screens were thought to be more beneficial and influence expectations. In addition, patients needed to have access to a suitable environment and equipment in order to undergo virtual rehabilitation.’ (page 13, infrastructure para)
9. (Page 12, Line 40) The situation that the patient is orientated to can provide capacity. – This sentence does not make sense to me.	We have changed to ‘The healthcare system can provide capacity.’
Discussion 1. The model (figure 1) explaining the relationship between the factors is complicated. The description is also complicated and repetitive. I think you could explain this much more simply.	This section has been rewritten. In addition, we have added Table 3 which offers practical questions to support the formation of preferences for patients. (page 16)
2. (Page 14) The capacity and available resources of a patient may influence their expectations of care. – You have defined available resources to be part of capacity so available resources is not needed in this sentence.	Thank you, we have removed this.
3. (Page 15) For some patients the physical attendance at a clinic was burdensome, yet they	We have reworded to:

preferred to attend due for a particular experience of care; for example to receive hands on manipulation – Could this be simplified?	'Some patients find the process of F2F attendance burdensome. Despite this, some of these patients preferred to receive hands on manipulation.' (page 17, para 2 below table 3)
4. (Page 16) Flexible options for patients and healthcare professionals to were provided to participate, both over phone and Skype as well as F2F. This sentence doesn't make sense. Could it be rephrased?	We have removed this sentence entirely as it is not a limitation.
5. (Page 16) Multiple factors were identified: The situation- 'The' should not be capitalised.	Many thanks for this, we have changed.
6. I would include a clear sentence highlighting the implications of your findings for clinical practice.	We have made this clearer with an amendment to a sentence that attempted to spell this out: 'This research illuminates the factors that appear to influence preference for patients. This is important for healthcare professionals; an understanding of preferences is essential to support the design of patient care pathways incorporating videoconferencing for consultations.' (page 19, conclusion para)

VERSION 2 – REVIEW

REVIEWER	Maria Larsson Dept of Health and Rehabilitation, Inst of neuroscience and physiology, The Sahlgrenska Academy at University of Gothenburg
REVIEW RETURNED	28-Oct-2020

GENERAL COMMENTS	bmjopen-2020-041038R.1 Thank you for the opportunity to review this revised version of the manuscript. I see that many of my comments have been met and others explained why they not have been met. Overall, I find the manuscript to be much improved in clarity. However, I still have some questions and comments. The overarching comment I have is still in regard to the aim: "To our knowledge, no studies have yet investigated the relationship between patient preferences around telemedicine services and their experience of burden of treatment. We need to better understand this to support the development of care pathways that take into account what offers patients increased utility. This paper therefore aims to identify, characterise, and explain factors that influence patient preferences for VCs in an orthopaedic rehabilitation setting."
--

	According to the title, this study describes what factors influence patient preference. But to better understand, increase utility and explain the factors, do we not need to know more about how these factors influence patient preferences? It might be that we come from different traditions of qualitative research and have different views on what to expect from the headings in the results section in regard to answering to the aim. If the themes and subthemes answering to the aim is just to find and describe what the factors are, I will accept this. But what I am trying to say is that I believe that the material could hold more to the results with a deeper analyse of the data and could provide headings with more meaning; to explain more of the meaning in the theme. We now are presented to the what but not to the how the factors influence the patient preference for VC. What in the (how did the) situation influence the patient preference for VC? How could you explain and understand how the factors influence patient preference, to better support the development of care pathways to offer patients increased utility, by providing more meaning to the themes? I do see that you characterise the sub-themes by describing more in text but, cannot the sub-theme and theme include something to characterise them there already? Is it not necessary for a theme or sub-theme to be just one or two words? I agree that you do not want the paper to be specific to orthopaedic rehabilitation consultations, but also to other forms of consultations within health care, I guess. The problem for me is that factors as situation, expectations, demands and capacity are such general concepts in themselves that they can be applied to any form of preferences, such as for food, car, school etc. This is why I would like something added to the general concept to be more specific to give meaning to this situation. Maybe, as I said before a different tradition but I find that maybe if you only added some of the explanation to the theme it would say much more, as for example: Instead of just Capacity - Capacity to allocate resources to care I appreciate the attempt to help the clinician to make use of the study's results (Table 3, which I think can be supplementary to the manuscript). Still, even as they relate to the themes, could not these questions have been generated without this qualitative study? For example based on the qualitative review or your own clinical experience? Thank you for providing the great looking and informative Figure 1 to better explain the process. As it explains that data matching phase 1 was temporarily disregarded I understand that there is substance to that your themes have been sorted into the similar clusters as your initial interview guide? (As this often is a problem in qualitative studies) Strategic and purposeful sampling I have heard of and, different ways to practice this but I am still not sure about what Maximum variation sampling means? I cannot see any reference to the sampling method? Does it mean that you have included informants with the largest differences? The oldest and the youngest? The ones with least and most experience from virtual consultation? Those with least and most clinical experience? I read that you have included different professionals, was this part of the sampling method? And what was the reason to use maximum variation sampling? Why is not Figure 2: model to illustrate interactions between mechanisms that influence preference for virtual consultations and the explanation of these relationships part of the results section?
--	---

	I am sorry to have left room for misunderstanding in regard to my comment on trustworthiness. What I meant was, to discuss you're the trustworthiness of your results to a greater extent, could be in terms of credibility, dependability, transferability of the results. Note: Abstract: Is there a comma missing in the first sentence? The manuscript might benefit from a thorough proof read, there seems for example as there is a comma missing in the first sentence (abstract) and I believe there are some 's missing. The sub-themes names have not been changed to care requirements and social demands in the text summerising the theme Demand. Communication technology is erased from title is but is still used at some places in the manuscript but changes at others.
--	---

REVIEWER	Dr Emma Phelps University of Oxford, UK
REVIEW RETURNED	03-Nov-2020

GENERAL COMMENTS	Thank you for the opportunity to review this revised manuscript. I think the authors have addressed the majority of comments well and the manuscript is improved. However, I believe addressing the following minor points would improve the manuscript further. Abstract  - The authors have added the following to the strengths and limitations "Theoretical insights and explanations generated from this paper are developed from empirical data" – is this correct for an abductive analysis? - "Single site qualitative study is not generalizable but mechanistic model is transportable between settings" I think you model is likely to be transferable between settings as it is very general but do you need to test this in some way before you can say this with such certainty? Could there be other factors in different clinical contexts which are not encompassed by your model? I would change this to "model is likely to be transferable". Method  - The authors mention maximum variation sampling. What do they mean by this as their recruitment method suggests opportunistic sampling? - The authors explain in Figure 1 and the text that data matching phase 1 is disregarded during the first stage of analysis and new themes are identified before the full analysis. Can they explain more about which of the results presented are new themes and which are from phase 1? Results  - At the end of theme one there is a short paragraph which links the situation to the other three themes but these themes have not yet been mentioned in the results. I would add a sentence to the beginning of the interview data section of the results explaining there are four themes: situation, expectations, demands and capacity. I would also move this paragraph to the beginning or end of the results as it seems out of place here. - In theme 3 the authors state: Healthcare professionals had an awareness of the potential limitations of VCs to offer empathy to the patients who desired it – I am not sure about the use of the word desire here. Empathy is an important part of the patient-clinician relationship and I think this suggests its only given because patients want it. I would rephrase "to offer empathy to the patients who need it" Discussion
--

	 - Second line of the discussion: These factors, empirically derived the study, were constructed from an abductive analysis- This does not make sense please rephrase - Please check section c of your model The relationship between Situation and Capacity: Patient capacity influences patient expectations directly via the demands and expectations of care. Do you mean patient capacity influences the situation indirectly via demands and expectations of care? - I do not understand what the following sentence means: The situation is firmly established once an equilibrium is reached between the situation and capacity. - In your discussion you state: A common theme in our data was the negative psychological impact some patients felt seeing themselves through a screen – I could not find this mentioned within your results - I think that it is good that you have now thought about the implications of your findings for clinical practice though the addition of table 3. However I wonder how clinicians might make use of them and whether they are feasible in this form for use in practice. Some of the questions are directed at patients while others are more relevant to clinicians considering the use of VCs. A decision aid for patients to help them consider their preferred consultation type F2F or VC that incorporates some of these questions might be a helpful way to make use of these findings for example.
--	---

VERSION 2 – AUTHOR RESPONSE

Comment	Response
Reviewer: 1 Reviewer Name: Maria Larsson Institution and Country: Dept of Health and Rehabilitation, Inst of neuroscience and physiology, The Sahlgrenska Academy at University of Gothenburg, Sweden	
The overarching comment I have is still in regard to the aim: “To our knowledge, no studies have yet investigated the relationship between patient preferences around telemedicine services and their experience of burden of treatment. We need to better understand this to support the development of care pathways that take into account what offers patients increased utility. This paper therefore aims to identify, characterise, and explain factors that influence patient preferences for VCs in an orthopaedic rehabilitation setting.” According to the title, this study describes what factors influence patient preference. But to better understand, increase utility and explain the factors, do we not need to know	We have changed the title to: Patient preferences for virtual consultations in an orthopaedic rehabilitation setting: a qualitative study as we think this is easier to understand. We have tried to write a tightly argued paper that is richly contextualised within the data. We believe that we have explained how these factors influence preference already within the discussion (under the headings a - e) and the model in figure 2 on page 14.

more about how these factors influence patient preferences?	
It might be that we come from different traditions of qualitative research and have different views on what to expect from the headings in the results section in regard to answering to the aim. If the themes and subthemes answering to the aim is just to find and describe what the factors are, I will accept this. But what I am trying to say is that I believe that the material could hold more to the results with a deeper analyse of the data and could provide headings with more meaning; to explain more of the meaning in the theme. We now are presented to the what but not to the how the factors influence the patient preference for VC. What in the (how did the) situation influence the patient preference for VC? How could you explain and understand how the factors influence patient preference, to better support the development of care pathways to offer patients increased utility, by providing more meaning to the themes? I do see that you characterise the sub-themes by describing more in text but, cannot the sub-theme and theme include something to characterisate them there already? Is it not necessary for a theme or sub-theme to be just one or two words?	The themes and subthemes are identifications and characterisations of these data. The explanatory aspect of the aims are situated within the discussion section on page 14 where we explicitly spell out how these mechanisms influence preference. We believe the aims of the paper have been achieved. It is possible that we do come from different traditions of qualitative research like you suggest. You have commented 'what I am trying to say is that I believe that the material could hold more to the results with a deeper analyse of the data' - In response to this we would like to point out that this paper is part of a larger body of work and there is a limit to what we can achieve in a single paper. This work is heading towards a theoretical model, but the theoretical model is developed from several sub studies as indicated in the paper: our systematic review (reference 15 in the submitted paper), this work, a choice experiment and an investigation of patients' perspectives on remote consultations that took place during service reconfiguration necessitated by the Covid-19 epidemic. We have made this clearer at the beginning of the methods section (page 4) and data analysis section (page 5). The theoretical model cannot depend on one single qualitative study. We therefore do not intend to stretch the analysis beyond what we have offered. We have made a change from 'capacity' to 'capacity to allocate resources to care' as suggested. In addition 'expectations' has been changed to 'expectations of care', 'demands' to 'demands on the patient', 'situation' to 'situation of care' and features from page 6.
I agree that you do not want the paper to be specific to orthopaedic rehabilitation consultations, but also to other forms of	We are using the terms we have chosen because they describe generalisable phenomena that are found in different contexts.

consultations within health care, I guess. The problem for me is that factors as situation, expectations, demands and capacity are such general concepts in themselves that they can be applied to any form of preferences, such as for food, car, school etc. This is why I would like something added to the general concept to be more specific to give meaning to this situation.	The characterisation of these factors are grounded from virtual consultations. We have made a change from 'capacity' to 'capacity to allocate resources to care' as suggested. In addition 'expectations' has been changed to 'expectations of care', 'demands' to 'demands on the patient', 'situation' to 'situation of care'.
Maybe, as I said before a different tradition but I find that maybe if you only added some of the explanation to the theme it would say much more, as for example: Instead of just Capacity - Capacity to allocate resources to care	As we have mentioned, the themes and subthemes have already been characterised in the text and in tables 2a-2d (from page 6). We have made a change from 'capacity' to 'capacity to allocate resources to care' as suggested. In addition, 'expectations' has been changed to 'expectations of care', 'demands' to 'demands on the patient', 'situation' to 'situation of care' and we hope this is clearer.
I appreciate the attempt to help the clinician to make use of the study's results (Table 3, which I think can be supplementary to the manuscript). Still, even as they relate to the themes, could not these questions have been generated without this qualitative study? For example, based on the qualitative review or your own clinical experience?	Thank you, Table 3 on page 16 was developed in response to your previous comments of clinical applicability. We spent a lot of time doing this, we feel it enhances the paper and therefore Table 3 should remain within the manuscript. As per reviewer 2's comments we have amended Table 3 slightly to include questions for patients and clinicians. The review papers were all offering different approaches with different underlying questions and in any case the data offered within the results, discussion and conclusion was limited – the results of this study are based on 44 qualitative interviews. As you suggest, it is possible these questions in table 3 could have been developed based on clinical experience. However, we would like to point out that these results are grounded in the empirical data in a manner outlined within the methods section of this paper – the questions generated in Table 3 have been developed from

	these results. Hopefully this brings with it more credibility to our results than them being based on our own clinical experience. We have made this clearer in page 15.
Thank you for providing the great looking and informative Figure 1 to better explain the process. As it explains that data matching phase 1 was temporarily disregarded I understand that there is substance to that your themes have been sorted into the similar clusters as your initial interview guide? (As this often is a problem in qualitative studies)	The qualitative interview schedule (in the appendix) was informed by the review. The themes from this study were identified following from a taxonomy and organised from there. Situation of care, expectations of care and capacity to allocate resources to care don't feature in the phase 1 model, although demands of care are discussed in the form of patient work within our previous paper (ref Gilbert, A.W., Jones, J., Jaggi, A. and May, C.R., 2020. Use of virtual consultations in an orthopaedic rehabilitation setting: how do changes in the work of being a patient influence patient preferences? A systematic review and qualitative synthesis. BMJ open, 10(9), p.e036197). We have added this into para 2 'results in context' on page 18. Our themes have not been sorted into similar clusters.
Strategic and purposeful sampling I have heard of and, different ways to practice this but I am still not sure about what Maximum variation sampling means? I cannot see any reference to the sampling method? Does it mean that you have included informants with the largest differences? The oldest and the youngest? The ones with least and most experience from virtual consultation? Those with least and most clinical experience? I read that you have included different professionals, was this part of the sampling method? And what was the reason to use maximum variation sampling?	We have sampled our patients on a set criteria of variation [set for age and gender] and we included a range of participants for gender and age (10 males, 5 under 50, 5 above; 12 females, 6 under 50 6 above). We wanted a wide variety of people. We were able to do this as we could select from incoming patients and our pool of clinicians. This has been spelt out within the manuscript on page 4 (in 'participants para). Virtual consultations were not being offered so least and most experience with virtual consultation is not relevant.
Why is not Figure 2: model to illustrate interactions between mechanisms that influence preference for virtual consultations and the	It is a discussion and interpretation of the results and we feel this sits best within the discussion section.

explanation of these relationships part of the results section?	
I am sorry to have left room for misunderstanding in regard to my comment on trustworthiness. What I meant was, to discuss you're the trustworthiness of your results to a greater extent, could be in terms of credibility, dependability, transferability of the results.	When we come to the theoretical paper for the project these are things we will have to discuss, there is not room for it in this paper. Your points are important and will be a core component of the thesis and the theoretical paper. We are already pushing the word limit for this paper and cannot fit all this in, as well as the additional things we have included.
Abstract: Is there a comma missing in the first sentence? The manuscript might benefit from a thorough proof read, there seems for example as there is a comma missing in the first sentence (abstract) and I believe there are some 's missing. The sub-themes names have not been changed to care requirements and social demands in the text summerising the theme Demand. Communication technology is erased from title is but is still used at some places in the manuscript but changes at others.	Yes, this has been changed to 'To identify, characterize and explain factors that influence patient preferences, from the perspective of patients and clinicians, for virtual consultations in an orthopaedic rehabilitation setting.' We have reflected changes in the text summarizing the theme 'demands on the patient' on page 10. Communication technology has been changed in the first para in the introduction (pg 3) and the data analysis section (pg 5). The term we will use is virtual consultations.
Reviewer: 2 Reviewer Name: Dr Emma Phelps Institution and Country: University of Oxford, UK	
Abstract - The authors have added the following to the strengths and limitations "Theoretical insights and explanations generated from this paper are developed from empirical data" – is this correct for an abductive analysis?	Yes, this is how we understand chapters 3 and 4 from Tavory and Timmermans (Tavory, I. and Timmermans, S., 2014. Abductive analysis: Theorizing qualitative research. University of Chicago Press). This interpretation informed our analysis. We accept there are differences in interpretations, but this interpretation has informed our analysis.
"Single site qualitative study is not generalizable but mechanistic model is transportable between settings" I think you model is likely to be transferable between settings as it is very general but do you need to test this in some way before you can say this with such certainty? Could there be other factors in different clinical contexts which are not encompassed by your	Changed to - "model is likely to be transferable" as suggested. Thank you for this, we completely agree. (page 2, strengths and limitations section)

model? I would change this to "model is likely to be transferable".	
Method  - The authors mention maximum variation sampling. What do they mean by this as their recruitment method suggests opportunistic sampling? 	It was not opportunistic as that implies it was anyone that came along. It was a deliberate, planned, systematic exercise where we sampled our patients on a set criteria of variation [set for age and gender] and we included a range of gender and age (10 males, 5 under 50, 5 above; 12 females, 6 under 50 6 above). We were able to do this as we could select from incoming patients. This has been spelt out within the manuscript on page 4 (participants para).
 - The authors explain in Figure 1 and the text that data matching phase 1 is disregarded during the first stage of analysis and new themes are identified before the full analysis. Can they explain more about which of the results presented are new themes and which are from phase 1? 	The themes from this study were identified following from a taxonomy and organised from there. Situation, expectations and capacity don't feature in the phase 1 model, although demands are discussed in the form of patient work. We have briefly mentioned this on page 18 (para 2): 'This current paper extends our previous model of patient preferences adding in: the situation of care, patient's expectations of care and patients ability to allocate resources to care (see Figure 2).'
Results  - At the end of theme one there is a short paragraph which links the situation to the other three themes but these themes have not yet been mentioned in the results. I would add a sentence to the beginning of the interview data section of the results explaining there are four themes: situation, expectations, demands and capacity. I would also move this paragraph to the beginning or end of the results as it seems out of place here. 	The initial para under 'interview data' on page 6 now reads as 'Four themes were identified from the data: (i) the situation of care, (ii) expectations of care, (iii) demands on the patient and (iv) capacity to allocate resources to care. Results from interviews are presented by theme and evidenced in tables 2a-2d which present data from both patients and healthcare professionals.' Paragraph in question 'The Situation is a factor that influences preference. Each situation is unique to the individual based on their clinical status, treatment requirements and the availability of care. The situation is influenced by the Capacity of the patient which in turn influences the Demands and the Expectations of patients. Whilst certain factors influence preferences for a patient in one direction, other factors may have an opposite effect.' Has been moved to the end of the results section on page 12.

- In theme 3 the authors state: Healthcare professionals had an awareness of the potential limitations of VCs to offer empathy to the patients who desired it – I am not sure about the use of the word desire here. Empathy is an important part of the patient-clinician relationship and I think this suggests its only given because patients want it. I would rephrase “to offer empathy to the patients who need it”	We think the word ‘desire’ is a reasonable word. People want something [empathy] and they desired it. Desire is about the expression of preference... We feel that ‘desires’ is the correct term here in the context of empathy. To make it clearer we have reworded to ‘Healthcare professionals had an awareness of the potential limitations to offer empathy via VC to the patients who desired it’ (page 8 – psychological status para)
Discussion - Second line of the discussion: These factors, empirically derived the study, were constructed from an abductive analysis- This does not make sense please rephrase	We have changed ‘to ‘These factors have been empirically derived.’ (page 15)
- Please check section c of your model The relationship between Situation and Capacity: Patient capacity influences patient expectations directly via the demands and expectations of care. Do you mean patient capacity influences the situation indirectly via demands and expectations of care?	In answer to your question: ‘Do you mean patient capacity influences the situation indirectly via demands and expectations of care?’ – yes. We have changed this. Thank you for highlighting. (page 14)
- I do not understand what the following sentence means: The situation is firmly established once an equilibrium is reached between the situation and capacity.	We have taken this sentence out. (page 15)
- In your discussion you state: A common theme in our data was the negative psychological impact some patients felt seeing themselves through a screen – I could not find this mentioned within your results	We have added the following quote ‘I guess because I was in a leg brace for so long, stuff did get shouted at me and I did get called things and that, so my self-confidence isn’t the best in the world [...] So to see myself in the corner of a screen doing something, it would stress me out for quite a huge amount of time. [P5]’ (page 8) Also the following sentence: ‘Some patients, however, found the idea of seeing themselves on a screen stressful’ (page 8 – psychological status para)
- I think that it is good that you have now thought about the implications of your findings for clinical practice though the addition of table	We have separated the questions into what patients might consider and what clinicians can ask patients in table 3. We have developed and

3. However I wonder how clinicians might make use of them and whether they are feasible in this form for use in practice. Some of the questions are directed at patients while others are more relevant to clinicians considering the use of VCs. A decision aid for patients to help them consider their preferred consultation type F2F or VC that incorporates some of these questions might be a helpful way to make use of these findings for example.	presented illustrations in images 1 – 4 for patients and clinicians to think about in the supplementary materials based on the patient questions. This is one way we are feeding back results of the study to participants. We agree, a decision aid might be very valuable. To create a decision aid is a huge amount and a different type of work from what is at play here.
--	--

VERSION 3 – REVIEW

REVIEWER	Maria Larsson Inst of neuroscience and physiology, Sahlgrenska Academy, University of Gothenburg, Sweden
REVIEW RETURNED	23-Dec-2020

GENERAL COMMENTS	Thank you for the revised version of the paper bmjopen-2020-041328 Virtual consultations have increased enormously during this year due to the pandemic situation. Even before the current situation virtual consultations were increasingly used. This submitted paper is therefore of importance to provide knowledge on how and for whom it would be of best use. The objective to identify, characterize and explain factors that influence patient preferences, from the perspective of patients and clinicians, for virtual consultations in an orthopaedic rehabilitation setting is therefore highly relevant. Unfortunately, I do not seem to be able to make my point fully understood when I tries to explain how I would like to see the themes and sub-themes include more meaning to be more self-explanatory. As for example, if the theme had a more self-explanatory label it would not be needed to have the explanation in parenthesis afterwards as in the abstract. For me, now the material is sorted into a description of factors related to influence preferences. But not until you read through all the text you get an explanation on how these can influence the preference. If the theme and subtheme itself had a higher degree of meaning it would guide the reader to easier catch the results. The tables are helpful, but still, I believe that the paper could have provided more information on what and how the factors influenced the preference for F2F vs VC. However, the authors explain that the paper is part of a larger body of work and we might have to be satisfied with the provision and description of factors which are of importance to form preferences and, the provision of questions, generated from this empirical data, to support identification of preference. As the patients' preferences for virtual consultations are not explicitly stated in the paper, I do not think the title should be changed to its' present form. I should include "factors influence".
---

	As a conceptual model is suggested, maybe a grounded theory approach would have been more beneficial for the material? To me it is confusing that the model is not part of results but of conclusion. The themes have not been changed in main documents Conclusions. Even though the paper has extended in words I still believe that trustworthiness of the study should be discussed, but I leave this decision to the Editor. Even though I understand that my comments can be perceived as rather critical I do believe that the study provides to the body of knowledge for use of virtual consultations and, it will be of interest to see the results of the CONNECT project in total.
--	---

VERSION 3 – AUTHOR RESPONSE

Reviewer: 1 Dr. Maria Larsson, Institute of Neuroscience and Physiology, Sahlgrenska Academy, University of Gothenburg Comments to the Author: Thank you for the revised version of the paper bmjopen-2020-041328 Virtual consultations have increased enormously during this year due to the pandemic situation. Even before the current situation virtual consultations were increasingly used. This submitted paper is therefore of importance to provide knowledge on how and for whom it would be of best use. The objective to identify, characterize and explain factors that influence patient preferences, from the perspective of patients and clinicians, for virtual consultations in an orthopaedic rehabilitation setting is therefore highly relevant.	Thank you for your comments.
Unfortunately, I do not seem to be able to make my point fully understood when I tries to explain how I would like to see the themes and sub-themes include more meaning to be more self-explanatory. As for example, if the theme had a more self-explanatory label it would not be needed to have the explanation in parenthesis afterwards as in the abstract. For me, now the material is sorted into a description of factors related to influence preferences. But not until you read through all the text you get an explanation on how these can influence the preference. If the theme and subtheme itself	We feel that it is not possible to provide a self-explanatory label without the parentheses and have left this unchanged.

had a higher degree of meaning it would guide the reader to easier catch the results.	
The tables are helpful, but still, I believe that the paper could have provided more information on what and how the factors influenced the preference for F2F vs VC. However, the authors explain that the paper is part of a larger body of work and we might have to be satisfied with the provision and description of factors which are of importance to form preferences and, the provision of questions, generated from this empirical data, to support identification of preference.	We have offered, in the model, an explanation as to how these factors influence preference. As you have mentioned, we have built on our previous work and will continue to build on this work. We have nothing further to add to this and hope that the editor deems our response to be satisfactory.
As the patients' preferences for virtual consultations are not explicitly stated in the paper, I do not think the title should be changed to its' present form. I should include "factors influence".	Thank you. We have changed the title to: 'Factors that influence patient preferences for virtual consultations in an orthopaedic rehabilitation setting: a qualitative study'
As a conceptual model is suggested, maybe a grounded theory approach would have been more beneficial for the material? To me it is confusing that the model is not part of results but of conclusion.	Abductive analysis is a highly suitable epistemological and methodological approach for theory generation and has been chosen as it addresses some of the weaknesses of grounded theory. As mentioned in the previous round of comments we are using the interpretation offered in Tavory and Timmermans for theory generation which attempts to move theory generation on from the 'inductive' nature of grounded theory. It is the position of the researcher that an 'abductive' approach (considering findings of our previous research) is a highly suitable way to address the phenomenon of interest. As we have previously pointed out, the model is a discussion and interpretation of the results and we feel this sits best within the discussion section.
The themes have not been changed in main documents Conclusions.	The themes have been updated in the conclusions

Even though the paper has extended in words I still believe that trustworthiness of the study should be discussed, but I leave this decision to the Editor.	As per the editor's advice on the matter, we have not changed this.
Even though I understand that my comments can be perceived as rather critical I do believe that the study provides to the body of knowledge for use of virtual consultations and, it will be of interest to see the results of the CONNECT project in total.	We thank you for your comments. These have been appropriately critical and welcomed and have influenced the paper, for which we are grateful.